# Increased Intake of Both Caffeine and Non-Caffeine Coffee Components Is Associated with Reduced NAFLD Severity in Subjects with Type 2 Diabetes

**DOI:** 10.3390/nu15010004

**Published:** 2022-12-20

**Authors:** Margarida Coelho, Rita S. Patarrão, Inês Sousa-Lima, Rogério T. Ribeiro, Maria João Meneses, Rita Andrade, Vera M. Mendes, Bruno Manadas, João Filipe Raposo, M. Paula Macedo, John G. Jones

**Affiliations:** 1CNC-Center for Neuroscience and Cell Biology, University of Coimbra, 3004-504 Coimbra, Portugal; 2CIBB-Centre for Innovative Biomedicine and Biotechnology, University of Coimbra, 3004-531 Coimbra, Portugal; 3III Institute for Interdisciplinary Research, University of Coimbra (IIIUC), 3030-789 Coimbra, Portugal; 4iNOVA4Health, NOVA Medical School-Faculdade de Ciências Médicas, NMS-FCM, Universidade Nova de Lisboa, 1169-056 Lisbon, Portugal; 5APDP-Diabetes Portugal, Education and Research Center, 1250-189 Lisbon, Portugal

**Keywords:** non-alcoholic fatty liver disease, fatty liver index, type 2 diabetes, caffeine, fibrosis

## Abstract

Coffee may protect against non-alcoholic fatty liver disease (NAFLD), but the roles of the caffeine and non-caffeine components are unclear. Coffee intake by 156 overweight subjects (87% with Type-2-Diabetes, T2D) was assessed via a questionnaire, with 98 subjects (all T2D) also providing a 24 h urine sample for quantification of coffee metabolites by LC–MS/MS. NAFLD was characterized by the fatty liver index (FLI) and by Fibroscan^®^ assessment of fibrosis. No associations were found between self-reported coffee intake and NAFLD parameters; however, total urine caffeine metabolites, defined as Σ_caffeine_ (caffeine + paraxanthine + theophylline), and adjusted for fat-free body mass, were significantly higher for subjects with no liver fibrosis than for those with fibrosis. Total non-caffeine metabolites, defined as Σ_ncm_ (trigonelline + caffeic acid + *p*-coumaric acid), showed a significant negative association with the FLI. Multiple regression analyses for overweight/obese T2D subjects (n = 89) showed that both Σ_caffeine_ and Σ_ncm_ were negatively associated with the FLI, after adjusting for age, sex, Hb_A1c_, ethanol intake and glomerular filtration rate. The theophylline fraction of Σ_caffeine_ was significantly increased with both fibrosis and the FLI, possibly reflecting elevated CYP2E1 activity—a hallmark of NAFLD worsening. Thus, for overweight/obese T2D patients, higher intake of both caffeine and non-caffeine coffee components is associated with less severe NAFLD. Caffeine metabolites represent novel markers of NAFLD progression.

## 1. Introduction

As a result of societal changes in diet and lifestyle, there has been an increase in obesity rates, with the incidence of non-alcoholic fatty liver disease (NAFLD) rapidly increasing in many Western countries. NAFLD can ultimately progress to severe and irreversible conditions such as cirrhosis and hepatocellular carcinoma, which generate enormous societal and healthcare burdens. Recent observational studies suggest that coffee consumption protects against NAFLD development and progression [1,2].

To date, these studies have relied on either self-reported information via questionnaires or interviews to evaluate coffee consumption. Aside from uncertainties arising from possible bias and/or misclassification in self-reporting, these data do not inform the intake of different coffee components, such as caffeine and polyphenols: this may be important in explaining the relationship between coffee intake and NAFLD, as both caffeine and non-caffeine metabolites may contribute to protection against NAFLD through independent mechanisms. Caffeine intake has been shown to be associated with decreased liver fibrosis in NAFLD and other chronic liver disease settings [3,4,5,6]. This effect may in part be mediated by inhibition of adenosine A_2A_ receptor-mediated induction of collagen production in hepatic stellate cells [4,7,8]. Adenosine A_2A_ and A_2B_ receptor signaling also contribute to regulating β-cell insulin secretion, as well as influencing insulin sensitivity in several tissues, including the liver [9,10], but the effects of caffeine on these parameters in relation to NAFLD are currently not known. In preclinical studies, other coffee components, such as trigonelline, chlorogenic acid and melanoidins have been shown to independently ameliorate NAFLD co-morbidities such as hepatic autophagy, oxidative stress, inflammation and intestinal barrier function [11,12,13,14,15], as well as improving systemic glucose homeostasis in both healthy and overweight subjects [16,17].

To date, almost all studies on the effects of coffee intake on human health have relied on self-reported coffee intake information. Given the inherent limitations and uncertainties of this approach, there have been efforts to obtain more defined and quantitative data on coffee intake, by measuring its major metabolites in urine collected over a 24 h interval [18,19]. With the exception of advanced cirrhosis and/or kidney dysfunction, coffee metabolites are cleared relatively quickly [20]: a 24 h urine sample, timed from before breakfast the preceding day to before breakfast on collection day, captures the majority of coffee metabolites that would be cleared over a typical day. To our knowledge, this approach has not yet been applied to study of the effects of coffee intake on markers of NAFLD severity. 

The effective dose of coffee metabolites seen by liver and other tissues not only depends on coffee intake, but also on the individual’s body mass. To date, correction for body mass has not been applied to any coffee intake study. Our study population had a wide range of body mass, with obesity also prevalent. As coffee metabolites are assumed to distribute predominantly in aqueous rather than lipid constituents, the amounts of recovered urinary coffee metabolites for each subject were adjusted for fat-free mass.

## 2. Materials and Methods

### 2.1. Materials

Caffeine (1 mg/mL), theobromine (0.1 mg/mL), theophylline (1 mg/mL), paraxanthine (1 mg/mL), ^13^C_3_-caffeine (1 mg/mL) stock solutions and theobromine-d6 (98% purity) were purchased from Sigma-Aldrich, St. Louis, MO, USA. Trigonelline, p-coumaric acid and catechin-2,3,4-^13^C_3_ in solid state were purchased from Sigma-Aldrich, while trans-caffeic acid was purchased from Fluka. Acetonitrile, methanol and water were LC–MS grade from VWR. Formic acid was LC–MS grade, and was purchased from Amresco.

### 2.2. Clinical Studies

Ethical permits were obtained from the Ethics Committee of Associação Protetora dos Diabéticos de Portugal (APDP). All subjects were volunteers, and written informed consent was obtained from all participants. The study protocol adhered to the Declaration of Helsinki. Subjects were recruited via the APDP clinic and, after providing informed consent, each subject was instructed to provide a 24 h urine sample, and to record their coffee and food intake over this period. The subjects were also requested to fill out a questionnaire on their customary consumption of beverages and caffeine-rich foods other than coffee, such as chocolate, based on that used by Modi et al. in their study of caffeine intake and liver fibrosis [21]. Within 24 h of finishing the urine collection, the subjects were admitted to the APDP clinic for anthropometric measurements, blood collection and assessment of liver steatosis and fibrosis with Fibroscan^®^. A subsection of subjects underwent the protocol without urine collection. Exclusion criteria included bowel disease or surgery, kidney disease, alcohol intake of >30 g per day for men and >20 g per day for women, and other chronic liver diseases, such as hepatitis A, B, C and Wilson’s disease.

### 2.3. Analysis of Urine Coffee Metabolites

Urine caffeine metabolites were analysed by LC–MS/MS, as described [22,23]. A similar protocol was developed for the non-caffeine metabolites. The detailed procedures are described in Appendix B.

### 2.4. Analysis of Blood Components, Body Composition and Fatty Liver Index

From the collected blood, plasma and serum were obtained. Clinical chemistry analyzers were used to assay glucose, HDL-cholesterol, LDL-cholesterol, total cholesterol, triglycerides, alanine aminotransferase (ALT), aspartate aminotransferase (AST) and γ-glutamyl transferase (GGT) (Beckman Coulter AU480), Hb_A1c_ (Menarini Hb 9210 Premier), and insulin (Architect Plus i1000SR), in accordance with standard operating procedures. Body composition, including fat-free body mass, was measured by bioimpedance, using a Body Composition Monitor (OMRON BF511). The fatty liver index (FLI) was calculated from triglyceride and GGT levels and from anthropometric parameters, according to the following formula [24]:FLI = (e^0.953 × ln (triglycerides) + 0.139 × BMI + 0.718 × ln (GGT) + 0.053 × waist circumference − 15.745^)/(1 + e^0.953 × ln (triglycerides) + 0.139 × BMI + 0.718 × ln (GGT) + 0.053 × waist circumference − 15.745^) × 100
Zonulin, a marker for intestinal barrier status [25,26], was measured from thawed serum samples with an ELISA kit (Immunodiagnostik AG, Bensheim, Germany), in accordance with the manufacturer’s instructions.

### 2.5. Alcohol Intake Measurement

Alcohol intake was calculated in terms of alcohol units per week, where 1 unit = 10 mL of pure ethanol. In the questionnaire information, it was designated that one small beer (200 mL, 5% alcohol) = 1 unit, one medium beer (330 mL, 5% alcohol) = 1.65 units, one large beer (500 mL, 5% alcohol) = 2.5 units, one glass of red or white wine (125 mL, 13% alcohol) = 1.625 units, and one measure of spirits (25 mL, 40% alcohol) = 1 unit. 

### 2.6. Data Analysis and Presentation

Statistical tests and correlation analyses were performed with Microsoft Excel and GraphPad Prism 8.0.1. Statistical significance was defined as a *p* value ≤ 0.05. Data were presented as means ± standard deviations in tables, and as box and whisker plots in graphs where, for a given data set, the box spanned the second and third quartile boundaries, with the horizontal line representing the median, and the whisker line spanned the maximum and minimum values.

## 3. Results

### 3.1. Study Cohort Characteristics

The study cohort was composed of almost equal proportions of overweight middle-aged males and females with high incidence of T2D (Table 1). The lipid profile was atherogenic, with above-normal triglyceride and total cholesterol levels, and with LDL-cholesterol well above the healthy range. The controlled attenuation parameter (CAP) and transient elastography (TE) measured by Fibroscan^®^ were consistent with moderate steatosis and mild fibrosis, respectively. Levels of circulating GGT, ALT and AST were at the upper limits of their normal ranges. The mean FLI score of 60 was at the cut-off value for ruling in NAFLD [24]. Mean alcohol intake was within the current 14 units per week guideline recommended by the NHS [27]. The subgroup of subjects that provided urine samples for determination of 24 h coffee metabolite amounts had identical characteristics, with the exception of T2D.

### 3.2. Urinary Coffee Metabolite Profile

The caffeine component of coffee can be metabolized via cytochrome P450 enzymes to paraxanthine, theophylline and theobromine (Figure 1). Since theobromine is also a constituent of tea and chocolate, it was excluded from the calculation of total caffeine metabolites. The remaining caffeine metabolites were dominated by paraxanthine, accounting for ~90% of the total, with caffeine and theophylline accounting for ~5–10% and ~1–5%, respectively (Appendix A). Of the non-caffeine components that were measured, trigonelline accounted for ~95% of the total, with caffeic acid and *p*-coumaric acid contributing about 4% and 1%, respectively. For both caffeine and non-caffeine metabolites, there was a robust association between their 24 h urine amounts and reported coffee consumption, as shown in Figure 2. Individual metabolites also showed a strong association with reported intake, with the exception of *p*-coumaric acid (Appendix A).

### 3.3. Coffee Metabolites and NAFLD Severity

To determine if there was a relationship between coffee metabolite output and NAFLD profile for this population, we examined the correlations between total urinary caffeine and non-caffeine metabolite levels, adjusted for fat-free mass, and three parameters associated with NAFLD: the FLI score, and fibrosis and steatosis readouts from Fibroscan^®^ (Table 2). There were no significant correlations of total urinary caffeine metabolites with any of the NAFLD parameters, or with zonulin, a serum marker of intestinal barrier function [28]. The total non-caffeine metabolites showed a significant negative correlation with the FLI but not with any other NAFLD parameters, nor with zonulin. We did not find any significant associations between self-reported coffee consumption, either adjusted or non-adjusted for fat-free mass, and any NAFLD parameter, either for the cohort that provided the 24 h urine samples or for the entire study population (Appendix A).

As the FLI, steatosis and fibrosis scores were each categorized according to NAFLD severity, we also examined the relationships between coffee metabolite amounts adjusted for fat-free mass and the different NAFLD categories (Figure 3). There were no significant differences between either total caffeine or total non-caffeine metabolites for subjects with (S1 + S2 + S3) or without (S0) steatosis, or for subjects where NAFLD could be ruled out (FLI < 30) vs. those where NAFLD was probable (FLI ≥ 30). Total caffeine metabolites were significantly higher for subjects with no fibrosis (F0), compared to those with fibrosis (F1–F4). There was no significant difference in total non-caffeine metabolites between these categories. For self-reported coffee intake adjusted for fat-free mass, no significant differences were found between any of the NAFLD categories (Appendix A).

To account for possible confounding factors in the relationship between coffee metabolites and FLI scores, a multiple regression analysis was performed. As the body mass index (BMI) is an important confounder of NAFLD, but is not a variable independent from the FLI (because the BMI is a component of the FLI formula), the subjects were stratified according to normal weight (BMI < 25 kg/m^2^) and overweight and obese (BMI ≥ 25 kg/m^2^). For overweight and obese T2D subjects (n = 86), both total caffeine metabolites and total non-caffeine metabolites retained a significant negative correlation with FLI scores after adjusting for age, sex, ethanol intake, glomerular filtration rate (GFR) and glycated hemoglobin (Hb_A1c_)—a marker of T2D severity (Appendix A). The same relationships between coffee metabolites and NAFLD parameters were also found when metabolite levels were normalized to total body mass (Appendix A).

### 3.4. Caffeine Metabolite Profiles and NAFLD Status

For the measured secondary metabolites of caffeine (paraxanthine and theophylline), there was a significant positive relationship between the fraction of caffeine metabolites accounted for by theophylline and some parameters of NAFLD severity (Figure 4). The theophylline fraction was significantly higher for subjects with either mild or moderate/severe fibrosis, relative to no fibrosis. It was also significantly higher for subjects with FLI scores ruling in vs. FLI scores ruling out NAFLD. The ratio of theophylline to paraxanthine also followed the same pattern (Appendix A). There was no relationship between the proportion of caffeine metabolites and steatosis severity. 

## 4. Discussion

### 4.1. Characterizing Coffee Consumption by 24 h Urine Analysis versus Self-Reporting

To date, most studies on the effects of coffee intake on the status of non-communicable chronic diseases, such as NAFLD and T2D, have relied on self-reported coffee intake; however, within the last few years, there has been an increasing shift to the analysis of coffee metabolites in plasma, serum and urine: this has been facilitated by the development of sensitive and robust LC– and UPLC–MS/MS methods for quantifying coffee metabolites in biofluids such as serum, plasma and cerebrospinal fluid [22,23,29], urine [30,31] and even fingertip sweat [32]. In five out of six measured urinary coffee metabolites (*p*-coumaric acid being the exception), we found robust and significant Pearson correlations, in the range of 0.29–0.43, with reported coffee intake (Appendix A). This is consistent with other studies that looked at the association between self-reported coffee intake data and urinary caffeine metabolites. Vanderlee et al. reported similar Pearson correlations of urinary caffeine, paraxanthine and theophylline concentrations with self-reported coffee intake, for 79 healthy young adults [19], while Petrovic et al. also reported strong associations between self-reported coffee intake and these three metabolites, but not with theobromine [18]. Of the non-caffeine metabolites, trigonelline was found to have a robust correlation with reported coffee intake for a group of 39 healthy subjects [33]. As for the Southern Italian population studied by Anty et al. [34], coffee is habitually drunk as espresso in Portugal. This may have a significant impact on the amount of caffeine and non-caffeine metabolites that are extracted, in comparison to coffee brewed over longer periods with higher volumes of water [35,36]. 

In previous studies, urinary coffee metabolites were not adjusted for either total body mass or fat-free mass. In the current study, we applied both 24 h urinary coffee metabolite and self-reported coffee intake data, to look for associations between coffee intake and NAFLD parameters in T2D subjects. While coffee metabolites were shown to have modest but significant associations with NAFLD parameters (Figure 3), none were found with self-reported coffee intake data (Appendix A). On this basis, the analysis of urinary coffee metabolites is a more precise and informative approach for studying the effects of coffee intake on NAFLD status, in comparison to questionnaire information. 

### 4.2. Coffee Intake and NAFLD Parameters

To date there have been several studies that have looked at the relationship between self-reported coffee intake and NAFLD status. In a cohort of subjects with NASH, coffee consumption was associated with a reduction in fibrosis risk [3]. Bambha et al. found that coffee consumption was associated with a lower risk of severe fibrosis in NAFLD patients that had preserved insulin sensitivity, but this effect was absent for insulin-resistant NAFLD subjects [37]. Veronese et al. did not find any association between coffee intake and the degree of steatosis in a Southern Italian population [38]. Anty et al. found that regular coffee, but not espresso, was protective against fibrosis in a group of morbidly obese European women [34]. Recent meta-analyses show a consensus that coffee intake protects against fibrosis for subjects with NAFLD [6,39,40,41,42,43], but there is a lack of agreement on whether coffee intake has any effect on NAFLD incidence in the general population, with three of these studies supporting an effect [40,42,43] and the other three concluding that there is no effect [6,39,41]. Our study found a significantly higher level of caffeine metabolites in subjects without fibrosis compared to those with fibrosis (Figure 3), but failed to find significant correlation between coffee metabolites and fibrosis, when studied as a continuous variable (Table 2). 

In addition to anthropometric and lipid inputs, the FLI has a liver enzyme (GGT) component. We found that total non-caffeine metabolites were negatively associated with FLI scores (Table 2), and that for overweight/obese T2D subjects, both caffeine and non-caffeine metabolites were associated with better (i.e., lower) FLI scores after adjusting for potential confounders (Appendix A). Xiao et al. also found an inverse relationship between coffee intake (including decaffeinated coffee) and circulating liver enzymes [44]. In subjects with liver cirrhosis, increased coffee consumption was related to lower prevalence of elevated aspartate and alanine aminotransferase levels [45].

### 4.3. Mechanisms by Which Coffee Metabolites May Protect against NAFLD

Coffee metabolites may protect against NAFLD development to more severe states. The risk factors for NAFLD and T2D are highly congruent, and include obesity, hypertriglyceridemia, insulin resistance and inflammation of both liver and adipose tissues [46,47,48]. More recently, intestinal microbiome dysbiosis and compromised intestinal barrier function have been implicated in disrupting hepatic nutrient metabolism via the generation of metabolites such as ethanol [49], through altering bile acid homeostasis [50,51], and by promoting visceral inflammation through the leakage of pro-inflammatory factors, such as endotoxin into the portal circulation [52]. 

To date, the best characterized direct effect of coffee on liver health is the attenuation of liver fibrosis by caffeine through its actions of antagonizing adenosine A_2A_ and A1 receptors in hepatic stellate cells, thereby interrupting signaling pathways for collagen production [4,8] as well as vasoconstriction and inflammation [53]. The development of fibrosis and inflammation are key factors in the progression of benign NAFLD towards more irreversible and severe states, such as NASH and cirrhosis. Additionally, there are also a number of extra-hepatic actions by both caffeine and non-caffeine metabolites that oppose NAFLD progression. In preclinical models of NAFLD, de-caffeinated coffee was shown to attenuate the loss of intestinal barrier function [12,13], which was associated with a less severe NAFLD phenotype [12]. In our study, the status of intestinal integrity was assessed by measuring the levels of circulating zonulin. Previous studies have shown associations between zonulin and severity of fatty liver disease in overweight subjects [25,28,54], as well as in patients with magnetic resonance and biopsy-characterized NAFLD and NASH [55]. In our study, no association was found between serum zonulin levels and either caffeine or non-caffeine metabolites (Table 2). One possible explanation is that the NAFLD severity of our study cohort was relatively mild, as seen by circulating liver enzyme levels and fibrosis scores (Table 1). Hendy et al. found that amongst their NAFLD cohort, zonulin levels were significantly higher in subjects with NASH, compared to those with simple steatosis, whose values did not significantly differ from those of healthy controls [55]. To the extent that coffee intake associates with circulating zonulin through its effects on intestinal integrity, this might be easier to detect in subjects with more advanced NAFLD, where there is a clear perturbation of serum zonulin levels. In the setting of mild NAFLD, this does not rule out possible associations of coffee metabolites with other parameters of intestinal health, such as microbiome status [56] and gastrointestinal hormone secretion [16].

### 4.4. Caffeine Metabolite Profiles and CYP2E1

The liver has an array of cytochrome P-450 (CYP) enzymes that metabolize ethanol, eicosanoid fatty acids, and a wide range of xenobiotic compounds, including many drugs and plant-derived natural products [57]. Initially considered to be solely involved in the detoxification of xenobiotic molecules, CYP isoforms may have wider effects on liver metabolism, in part via their oxidation of physiologically active eicosanoids such as prostaglandins [57]. Caffeine is metabolized to paraxanthine via CYP1A2, while its conversion to theophylline or theobromine is mediated not only by this enzyme, but also by CYP2E1. Preclinical studies have shown that CYP2E1 expression is upregulated by chronic alcohol intake and by high fat intake [58,59,60], while biopsy studies have revealed upregulated CYP2E1 expression in livers of NAFLD patients, compared to those of healthy subjects [61]. Not only is increased CYP2E1 expression associated with NAFLD, but it is also implicated in its progression to more severe states, by contributing to increased hepatic oxidative stress [60,62]. Our observation of a significant positive correlation between the fraction of urinary caffeine metabolites represented by theophylline and FLI scores (Figure 4) is consistent with such upregulated CYP2E1 expression, and suggests that this measurement could be a useful non-invasive marker of hepatic CYP2E1 status. 

### 4.5. Study Limitations

Our study has several limitations that need to be taken into consideration. While analysis of coffee metabolites from 24 h urine samples may provide a more informative indication of coffee intake, compared to questionnaire data, it is nevertheless based on intake over a single day. Whether this interval is truly representative of the subject’s habitual coffee intake is not known. Moreover, the urine metabolites that were measured can be derived from foods and beverages other than coffee. Finally, our measurement of NAFLD involved the use of serum markers and ultrasound/elastography, all of which are less precise than liver biopsy and magnetic resonance spectroscopy/imaging modalities in NAFLD diagnosis and staging.

Authors should discuss the results and how they can be interpreted from the perspective of previous studies and of the working hypotheses. The findings and their implications should be discussed in the broadest context possible. Future research directions may also be highlighted.

## 5. Conclusions

There is a general consensus that coffee intake is associated with modest but significant protection against NAFLD. To date, this has been entirely based on questionnaire data that, among other things, does not provide information on the role of different coffee components in protecting against NAFLD. Our study indicates that higher cumulative amounts of both caffeine and non-caffeine metabolites measured in a 24 h urine collection are associated with a less severe NAFLD profile. Finally, the profile of urine caffeine metabolites is sensitive to NAFLD severity, and may serve as a non-invasive marker of hepatic CYP2E1 expression, an important driver of NAFLD progression.

## Figures and Tables

**Figure 1 nutrients-15-00004-f001:**
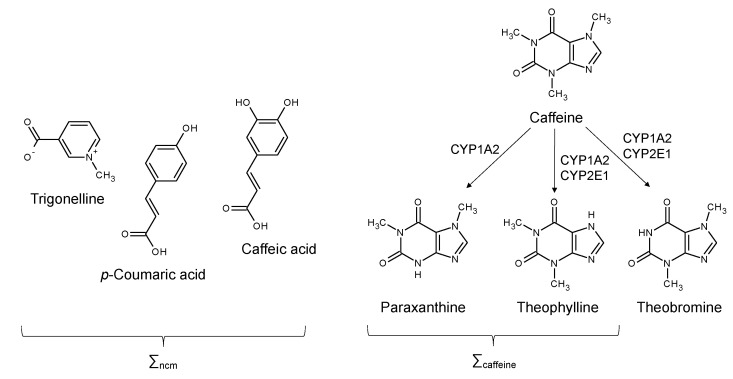
Caffeine metabolites that were analyzed in the subject urine samples, including caffeine and two of its secondary metabolites—paraxanthine and theophylline (∑_caffeine_)—and the non-caffeine metab-olites, trigonelline, p-coumaric acid and caffeic acid (∑_ncm_). Also shown are the cytochrome P450 mixed-function oxidases (CYP1A2 and CYP2E1) that convert caffeine to these secondary metabo-lites.

**Figure 2 nutrients-15-00004-f002:**
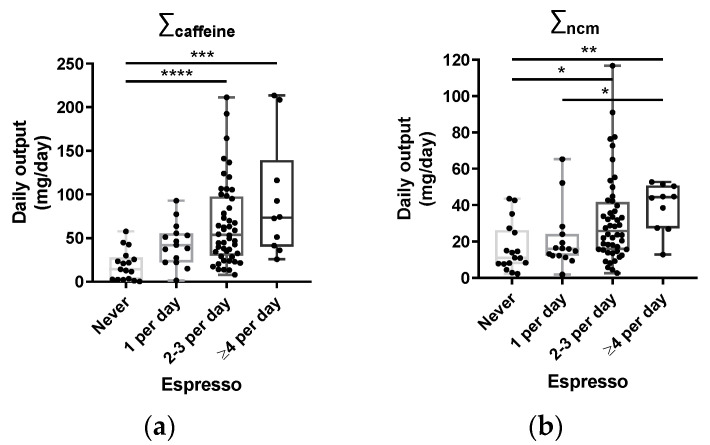
Reported coffee consumption associated with 24 h urine amounts of coffee components: (**a**) relationship between coffee metabolites measured in the 24 h urine samples and reported espresso intake for total caffeine metabolites (∑_caffeine_); (**b**) total non-caffeine metabolites (∑_ncm_). * = significant difference (*p* ≤ 0.05, Kruskal–Wallis test with Dunn’s multiple comparisons); ** = significant difference (*p* ≤ 0.01, Kruskal–Wallis test with Dunn’s multiple comparisons); *** = significant difference (*p* ≤ 0.001, Kruskal–Wallis test with Dunn’s multiple comparisons); **** = significant difference (*p* ≤ 0.0001, Kruskal–Wallis test with Dunn’s multiple comparisons).

**Figure 3 nutrients-15-00004-f003:**
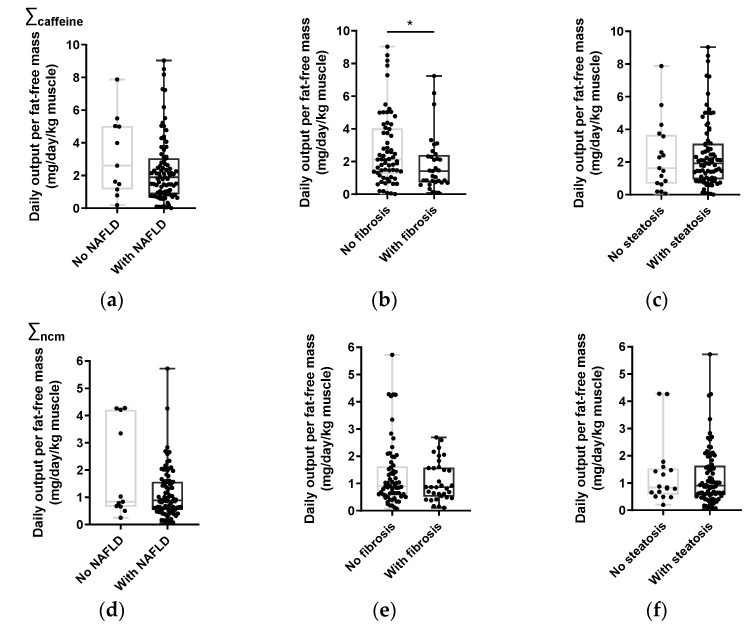
High total caffeine metabolites are associated with less fibrosis. Levels of total caffeine metabolites (**a**–**c**) and total non-caffeine metabolites (**d**–**f**), adjusted for fat-free mass with different categories of the fatty liver index (FLI) (**a**,**d**), scores of fibrosis (**b**,**e**) and steatosis (**c**,**f**). * = significant difference (*p* ≤ 0.05, Mann–Whitney test).

**Figure 4 nutrients-15-00004-f004:**
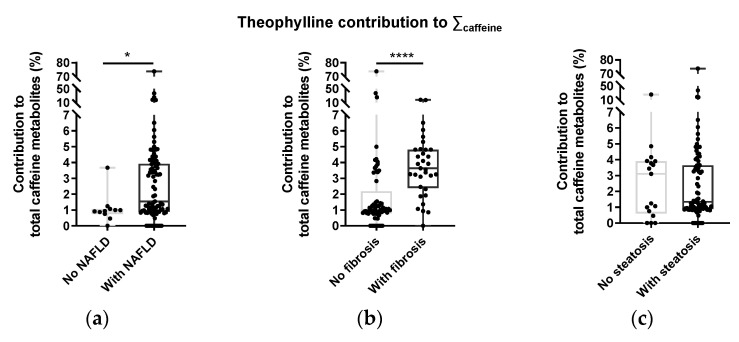
Theophylline was higher in subjects with NAFLD and fibrosis. Theophylline fraction of total caffeine metabolites and categories of the fatty liver index (FLI) (**a**) and scores of fibrosis (**b**) and steatosis (**c**). * = significant difference (p ≤ 0.05, Mann–Whitney test); **** = significant difference (*p* ≤ 0.0001, Mann–Whitney test).

**Table 1 nutrients-15-00004-t001:** Population characteristics. Characteristics of the total subject population and the subset of subjects that provided urine for analysis of coffee metabolites. Values are expressed as mean ± standard deviation.

Parameters	Values	
	Total (n = 156)	With Urine Data (n = 98)
Age (yr)	59 ± 9	60 ± 6
Sex distribution (F/M)	76/80	52/46
Type 2 Diabetes incidence & duration (yr)	135/156 (87%), 11 ± 7	98/98 (100%), 11 ± 6
Body mass index (BMI, kg·m^−2^)	29 ± 5	30 ± 4
Aspartate aminotransferase (AST, U·L^−1^)	23 ± 6	22 ± 4
Alanine aminotransferase (ALT, U·L^−1^)	27 ± 12	26 ± 7
γ-glutamyltransferase (GGT, U·L^−1^)	31 ± 21	31 ± 14
Fibrosis (transient elastography, kPa)	5.7 ± 2.9	6.1 ± 2.1
Steatosis (controlled attenuation parameter, dB·m^−1^)	283 ± 56	290 ± 44
Fatty liver index (FLI) score	60 ± 26	63 ± 20
Total Cholesterol (mg·dL^−1^)	180 ± 39	178 ± 32
HDL-Cholesterol (mg·dL^−1^)	51 ± 11	49 ± 9
LDL-Cholesterol (mg·dL^−1^)	121 ± 32	120 ± 27
Triglyceride (mg·dL^−1^)	141 ± 66	149 ± 53
Glucose (mg·dL^−1^)	145 ± 41	153 ± 31
Insulin (µU·mL^−1^)	11 ± 10	11 ± 7
Zonulin (ng·mL^−1^)	46 ± 10	48 ± 7
Alcohol intake (alcohol units per week)	8 ± 14	10 ± 16

Abbreviations: high-density lipoprotein (HDL); low-density lipoprotein (LDL).

**Table 2 nutrients-15-00004-t002:** Non-caffeine metabolites correlated with the FLI. Correlations, expressed as Pearson coefficients, of total urinary coffee metabolites [(caffeine metabolites (Σ_caffeine_) and non-caffeine metabolites (Σ_ncm_)], expressed per kg of fat-free mass, with the fatty liver index (FLI), fibrosis and steatosis parameters measured by Fibroscan^®^, and with serum zonulin levels. The *p* value for each Pearson coefficient is shown alongside in parentheses.

Liver Parameter	Correlation with Σ_caffeine_	Correlation with Σ_ncm_
FLI	−0.1783 (*p* = 0.0789)	−0.2266 * (*p* = 0.0249)
Fibrosis (kPa)	−0.1090 (p = 0.2852)	−0.0646 (*p* = 0.5271)
Steatosis (CAP, db·m^−1^)	−0.0734 (*p* = 0.4728)	−0.0860 (*p* = 0.3996)
Zonulin (ng·mL^−1^)	−0.0804 (*p* = 0.4510)	0.06918 (*p* = 0.5171)

* = significant difference (*p* ≤ 0.05, Pearson correlation). Abbreviations: Controlled attenuation parameter (CAP).

## Data Availability

The data presented in this study are available on request from the corresponding author. The data are not publicly available, due to restrictions presented in the informed consent, related to data storage.

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
