# Peer review of "Increased Intake of Both Caffeine and Non-Caffeine Coffee Components Is Associated with Reduced NAFLD Severity in Subjects with Type 2 Diabetes"

_nutrients, 2022, doi:10.3390/nu15010004_

Round 1

Reviewer 1 Report

This is a carefully planned study reporting on the relationship between caffeine metabolites and the parameters of NAFLD. Also highlighted are the issues related to self-reporting. Of course, that brings into question the accuracy or alcohol intake, but that does not diminish the importance of this paper.

Very nice paper.

Comments:

It is important to correct the misconception that caffeine “Protects” against NAFLD. It does not. Rather, it reduces the severity, or improves healthy parameters, or something like that.

Last sentence of Abstract…”inform”????

Fig. 1 is nice and informative. The authors are aware that CYP1A2 and CYP2E1 are not uniformly expressed in humans. There are some papers and reviews by Daniel Nebert on CYP1A distributions in humans. The authors may want to say a word or two on this subject in the ms. That is, similar caffeine intake may not result in similar metabolite profiles or time course. The second Fig 1 (Actually Fig. 2) sort of addresses that.

Author Response

Point 1: It is important to correct the misconception that caffeine “Protects” against NAFLD. It does not. Rather, it reduces the severity, or improves healthy parameters, or something like that.

Response 1: We have changed the title of the manuscript according to the reviewer suggestion to: “Increased intake of both caffeine and non-caffeine coffee components is associated with reduced NAFLD severity in subjects with type 2 diabetes”.

Point 2: Last sentence of Abstract…”inform”????

Response 2: We have altered the last sentence on the abstract.

Point 3: Fig. 1 is nice and informative. The authors are aware that CYP1A2 and CYP2E1 are not uniformly expressed in humans. There are some papers and reviews by Daniel Nebert on CYP1A distributions in humans. The authors may want to say a word or two on this subject in the ms. That is, similar caffeine intake may not result in similar metabolite profiles or time course. The second Fig 1 (Actually Fig. 2) sort of addresses that.

Response 3: Two sentences highlighting CYP role in the liver with a reference by the suggested author were added (lines 319-324). The typo on the figure legend for Figure 2 has also been corrected.

Reviewer 2 Report

The ameliorative effect of coffee on NAFLD has been reported in many papers. However, it has remained unclear as to what components are responsible for the effect. This paper describes a questionnaire-based study of coffee intake in 156 overwight patients and the quantification of caffeine components in the urine of 98 patients with type 2 diabetes mellitus.

In this study, there was no association between coffee intake and NAFLD parameters. However, it was found that for overweight/obese T2D patients, higher intake of both caffeine and non-caffeine coffee components was associated with less severe NAFLD.

This is an excellent study rich in further trials in the area of the coffee/NAFLD association.

Author Response

We thank the reviewer for these positive remarks.